# Two Decades of Brain Tumour Imaging with O-(2-[^18^F]fluoroethyl)-L-tyrosine PET: The Forschungszentrum Jülich Experience

**DOI:** 10.3390/cancers14143336

**Published:** 2022-07-08

**Authors:** Alexander Heinzel, Daniela Dedic, Norbert Galldiks, Philipp Lohmann, Gabriele Stoffels, Christian P. Filss, Martin Kocher, Filippo Migliorini, Kim N. H. Dillen, Stefanie Geisler, Carina Stegmayr, Antje Willuweit, Michael Sabel, Marion Rapp, Michael J. Eble, Marc Piroth, Hans Clusmann, Daniel Delev, Elena K. Bauer, Garry Ceccon, Veronika Dunkl, Jurij Rosen, Caroline Tscherpel, Jan-Michael Werner, Maximilian I. Ruge, Roland Goldbrunner, Jürgen Hampl, Carolin Weiss Lucas, Ulrich Herrlinger, Gabriele D. Maurer, Joachim P. Steinbach, Jörg Mauler, Wieland A. Worthoff, Bernd N. Neumaier, Christoph Lerche, Gereon R. Fink, Nadim Jon Shah, Felix M. Mottaghy, Karl-Josef Langen

**Affiliations:** 1Department of Nuclear Medicine, RWTH University Hospital, D-52074 Aachen, Germany; aheinzel@ukaachen.de (A.H.); danieladedic@aol.com (D.D.); c.filss@fz-juelich.de (C.P.F.); fmottaghy@ukaachen.de (F.M.M.); 2Institute of Neuroscience and Medicine (INM-3, INM-4, INM-5, INM-11), Forschungszentrum Jülich, D-52425 Jülich, Germany; n.galldiks@fz-juelich.de (N.G.); p.lohmann@fz-juelich.de (P.L.); g.stoffels@fz-juelich.de (G.S.); m.kocher@fz-juelich.de (M.K.); s.geisler@fz-juelich.de (S.G.); c.stegmayr@fz-juelich.de (C.S.); a.willuweit@fz-juelich.de (A.W.); j.mauler@fz-juelich.de (J.M.); w.worthoff@fz-juelich.de (W.A.W.); b.neumaier@fz-juelich.de (B.N.N.); c.lerche@fz-juelich.de (C.L.); g.r.fink@fz-juelich.de (G.R.F.); n.j.shah@fz-juelich.de (N.J.S.); 3Center of Integrated Oncology (CIO), University of Aachen, D-52074 Aachen, D-53127 Bonn, D-50937 Cologne, D-40225 Düsseldorf, Germany; michael.sabel@med.uni-duesseldorf.de (M.S.); marion.rapp@med.uni-duesseldorf.de (M.R.); meble@ukaachen.de (M.J.E.); hclusmann@ukaachen.de (H.C.); ddelev@ukaachen.de (D.D.); elena.bauer@uk-koeln.de (E.K.B.); garry.ceccon@uk-koeln.de (G.C.); veronika.dunkl@uk-koeln.de (V.D.); jurij.rosen@uk-koeln.de (J.R.); caroline.tscherpel@uk-koeln.de (C.T.); jan-michael.werner@uk-koeln.de (J.-M.W.); maximilian.ruge@uk-koeln.de (M.I.R.); roland.goldbrunner@uk-koeln.de (R.G.); juergen.hampl@uk-koeln.de (J.H.); carolin.weiss-lucas@uk-koeln.de (C.W.L.); ulrich.herrlinger@ukbonn.de (U.H.); 4Department of Neurology, University Hospital Cologne, D-50937 Cologne, Germany; 5Center for Neurosurgery, Department of Stereotactic and Functional Neurosurgery, University Hospital Cologne, D-50937 Cologne, Germany; 6Department of Orthopedics, Trauma, and Reconstructive Surgery, RWTH University Hospital, D-52074 Aachen, Germany; fmigliorini@ukaachen.de; 7Department of Palliative Medicine, University Hospital Cologne, D-50937 Cologne, Germany; kim.dillen1@uk-koeln.de; 8Department of Neurosurgery, University Hospital Düsseldorf, D-40225 Düsseldorf, Germany; 9Department of Radiotherapy, RWTH University Hospital, D-52074 Aachen, Germany; 10Department of Radiation Oncology, Helios University Hospital Wuppertal, Witten/Herdecke University, D-42283 Wuppertal, Germany; marc.piroth@helios-kliniken.de; 11Department of Neurosurgery, RWTH University Hospital, D-52074 Aachen, Germany; 12Center for Neurosurgery, Department of General Neurosurgery, University Hospital Cologne, D-50937 Cologne, Germany; 13Division of Clinical Neuro-Oncology, Department of Neurology, University of Bonn Medical Center, D-53127 Bonn, Germany; 14Dr. Senckenberg Institute of Neurooncology, University Cancer Center Frankfurt (UCT), Goethe University Hospital, D-60528 Frankfurt am Main, Germany; gabriele.maurer@pei.de (G.D.M.); joachim.steinbach@kgu.de (J.P.S.); 15Institute of Radiochemistry and Experimental Molecular Imaging, University Hospital Cologne, D-50937 Cologne, Germany; 16JARA—BRAIN—Translational Medicine, RWTH Aachen University, D-52074 Aachen, Germany; 17Department of Neurology, RWTH Aachen University Hospital, D-52074 Aachen, Germany; 18Department of Radiology and Nuclear Medicine, Maastricht University Medical Center (MUMC+), P.O. Box 5800, 6202 Maastricht, The Netherlands

**Keywords:** brain tumour diagnosis, positron emission tomography, O-(2-[^18^F]fluoroethyl)-L-tyrosine (FET), glioma, brain metastasis, amino acid PET

## Abstract

**Simple Summary:**

PET using radiolabelled amino acids has become an essential tool for diagnosing brain tumours in addition to MRI. O-(2-[^18^F]fluoroethyl)-L-tyrosine (FET) is one of the most successful tracers in the field. We analysed our database of 6534 FET PET examinations regarding the diagnostic needs and preferences of the referring physicians for FET PET in the clinical decision-making process. The demand for FET PET increased considerably in the last decade, especially for differentiating tumour progress from treatment-related changes in gliomas. Accordingly, referring physicians rated the diagnostics of recurrent glioma and recurrent brain metastases as the most relevant indication for FET PET. The analysis and survey results confirm the high relevance of FET PET in the clinical diagnosis of brain tumours and support the need for approval for routine use.

**Abstract:**

O-(2-[^18^F]fluoroethyl)-L-tyrosine (FET) is a widely used amino acid tracer for positron emission tomography (PET) imaging of brain tumours. This retrospective study and survey aimed to analyse our extensive database regarding the development of FET PET investigations, indications, and the referring physicians’ rating concerning the role of FET PET in the clinical decision-making process. Between 2006 and 2019, we performed 6534 FET PET scans on 3928 different patients against a backdrop of growing demand for FET PET. In 2019, indications for the use of FET PET were as follows: suspected recurrent glioma (46%), unclear brain lesions (20%), treatment monitoring (19%), and suspected recurrent brain metastasis (13%). The referring physicians were neurosurgeons (60%), neurologists (19%), radiation oncologists (11%), general oncologists (3%), and other physicians (7%). Most patients travelled 50 to 75 km, but 9% travelled more than 200 km. The role of FET PET in decision-making in clinical practice was evaluated by a questionnaire consisting of 30 questions, which was filled out by 23 referring physicians with long experience in FET PET. Fifty to seventy per cent rated FET PET as being important for different aspects of the assessment of newly diagnosed gliomas, including differential diagnosis, delineation of tumour extent for biopsy guidance, and treatment planning such as surgery or radiotherapy, 95% for the diagnosis of recurrent glioma, and 68% for the diagnosis of recurrent brain metastases. Approximately 50% of the referring physicians rated FET PET as necessary for treatment monitoring in patients with glioma or brain metastases. All referring physicians stated that the availability of FET PET is essential and that it should be approved for routine use. Although the present analysis is limited by the fact that only physicians who frequently referred patients for FET PET participated in the survey, the results confirm the high relevance of FET PET in the clinical diagnosis of brain tumours and support the need for its approval for routine use.

## 1. Introduction

Today, contrast-enhanced magnetic resonance imaging (MRI) is the method of choice for diagnosing brain tumours and clinical follow-up. Standard T1-weighted and T2-weighted sequences provide high-resolution structural imaging and enable a reliable diagnosis in many cases [1]. Nevertheless, differentiating tumour tissue from non-specific tissue changes can be problematic, especially in diffuse tumour growth, lack of contrast enhancement, and treatment-related tissue changes in pretreated tumours. Many studies have demonstrated that positron emission tomography (PET) using radiolabelled amino acids provides decisive additional information to solve the aforementioned problems [2]. Consequently, the Response Assessment in Neuro-Oncology (RANO) working group has recommended using amino acid PET imaging and MRI in all stages of brain tumour management [3,4,5].

In the 1990s, O-(2-[^18^F]fluoroethyl)-L-tyrosine (FET) was developed at the Forschungszentrum Jülich (FZJ) to provide an ^18^F-labelled amino acid PET tracer with a half-life (110 min) suitable for routine clinical applications compared to the shorter-lived carbon-11-labelled amino acid tracers (half-life: 20 min; e.g., L-[methyl-^11^C]methionine (MET)) [6,7,8,9]. In the last two decades, our team has contributed to the preclinical and clinical evaluation of FET as a tracer of amino acid transport in brain tumours by publishing more than 170 research papers. Fortunately, FET has become one of the most successful amino acid tracers for brain tumour imaging [10,11]. The high diagnostic value of FET PET has attracted clinical interest among neuro-oncologists, neurosurgeons, and radiation oncologists in the university clinics in our area, resulting in considerable growth in the number of investigated patients. Meanwhile, our department has performed more than 8000 FET PET investigations (May 2022) since 2001.

As several excellent review articles and meta-analyses have discussed the sensitivity, specificity, and diagnostic accuracy of FET PET in various aspects of brain tumour imaging, this study’s intention was not to review these aspects again [1,2,3,4,7,11,12,13,14].

Instead, this survey’s rationale was to obtain information on the acceptance and role of FET PET, based on the development of clinical demands and the needs of the neurosurgeon, neuro-oncologist, or radiation oncologist. Furthermore, we were interested in assessing FET PET’s necessity according to the referring doctors.

Until 2005, the FET PET examinations in our department were primarily conducted within the framework of prospective studies [15,16,17,18,19]. The considerable increase in the number of examinations after that, especially in the last ten years, predominantly reflects the clinical needs of the referring physicians (Figure 1). Therefore, we assumed that the database accumulated at the FZJ provides an excellent footing for such an analysis because our situation is different from PET centres located at university hospitals or other specialised institutions, where the decision to use FET PET might be influenced by local factors such as accessibility, preferences of the tumour therapists or the tumour board, or economic issues. For example, internal billing of the PET costs does not play a role at our institution as the FET PET investigations are free of charge for the referring clinic, and the effort of the referring physician is limited to the arrangement of an appointment. This unique arrangement at the FZJ provides a patient collective that reflects the diagnostic needs in neuro-oncology beyond conventional neuro-radiological assessment and for which FET PET plays an essential role in clinical decision making. Thus, this retrospective study and survey aimed to analyse the clinical relevance of FET PET based on the referral practice in a large population of patients driven by clinical needs. The results can contribute to the current debate on whether a broader availability of amino acid PET is necessary for the standard care of brain tumour patients.

## 2. Patients and Methods

### 2.1. Study Design

The present study was conducted following the recommendation of the Strengthening the Reporting of Observational Studies in Epidemiology: the STROBE statement [20] and includes our experiences with FET PET since its introduction in 2001. In the years from 2001 to 2005, patients were mainly examined in the context of prospective studies (n = 374), such as the diagnostic potential of FET PET to estimate the tumour extent in untreated gliomas in a biopsy-controlled study, whole-body distribution, radiation dosimetry, and in comparison with other tracers [18,21]. Since these studies do not reflect the clinical needs, these cohorts were excluded from the present evaluation. Therefore, in this retrospective study, only data from patients having a FET PET examination at the Institute of Neuroscience and Medicine of the FZJ from 2006 until the end of 2019 were evaluated. During this time, 6534 FET PET investigations were performed on 3928 patients, which served as this study’s database. The preparation of patients, the FET PET procedure, synthesis method, scanners used, reconstruction and correction methods, and the evaluation of the FET PET data are not the subject of this study and have been described in detail elsewhere [22,23,24].

### 2.2. Ethics Statement

The study adheres to the standards established in the Declaration of Helsinki. All patients provided written informed consent before each FET PET investigation. The ethical committee of the Medical Faculty of the RWTH Aachen University approved this retrospective analysis of patient data (EK 386/20). 

### 2.3. Objectives

We evaluated the database of FET PET examinations in the FZJ from 2006 to 2019 under the following aspects:Total number of examinations;Number of patients with multiple examinations;Development of the number of examinations over time;Indication for referral to FET PET;The referring physicians’ specialty;Distribution of FET PET examinations among the referring clinics;Distance of the referrer from the FZJ.

### 2.4. Survey

In the neuro-oncology centres with more than 100 referrals, 23 specialists with more than 3 years of experience in using FET PET in brain tumour diagnosis were identified. All of them completed a questionnaire designed for this study. This questionnaire consisted of 30 Likert-like questions, which aimed at evaluating the relevance of FET PET under the following aspects:Rating of the additional value of FET PET in comparison to conventional MRI in glioma patients at initial diagnosis, in early postoperative assessment, in the case of suspected tumour recurrence, and for therapy monitoring.Rating of the additional value of FET PET in comparison to conventional MRI in patients with brain metastases in the case of suspected tumour recurrence and for therapy monitoring.Percentage of patients in which FET PET is considered helpful for the various indications.Rating of the value of advanced MR procedures (e.g., PWI, MRS, and DWI) for the various indications compared to FET PET.General statements on the need for FET PET in brain tumour assessment.
Each observer rated each question or statement according to a rating scale ranging from 1, very important, to 5, unimportant, or in the case of percentages in five steps: <10%; 25%, 50%, 75%, and >90%.

Appendix A of the Appendix A provides the English version of the questionnaire.

### 2.5. Statistics

Descriptive statistics are provided as mean ± standard deviation (SD). One-way repeated measures ANOVAs with All Pairwise Multiple Comparison Procedures (Holm–Sidak method) were performed to determine differences across groups of medical specialists. P values of 0.05 or less were considered significant. Statistical analysis was performed using SigmaPlot for Windows, Version 12.5.

## 3. Results

### 3.1. Database

A total of 6534 FET PET studies were performed on 3928 patients between 2006 and the end of 2019. Figure 1 illustrates the growing demand for FET PET examinations. Sixty-six per cent of the patients had one PET scan, 18% had two PET scans, and the remaining patients had three or more PET scans. Predominantly, glioma patients were examined using FET PET, but from 2010 onwards, patients with brain metastases were increasingly referred, accounting for 13% of the patients examined in the last survey year.

Based on the referral spectrum of 2019, the frequencies of FET PET indications were: the diagnosis of suspected recurrent glioma in 46%, differential diagnosis of unclear brain lesions in 20%, treatment monitoring in patients with glioma in 19%, diagnosis of suspected recurrent brain metastasis in 13%, and 1% with other indications (Figure 2). Forty-three percent of the patients were treated at the University Clinic of Düsseldorf (distance from the FZJ, 58 km), 28% at the University Clinic of Cologne (distance from the FZJ, 48 km), 13% at the University Clinic of Aachen (distance from the FZJ, 37 km), 9% at the University Clinic of Frankfurt (distance from the FZJ, 233 km), and 7% at the Clinic and Gammaknife Center in Krefeld (distance from the FZJ, 70 km). Thus, most patients were referred by centres located up to 75 km away, but 9% also were referred from centres at a distance of more than 200 km. In addition, most patients for FET PET were referred by neurosurgeons (60%), followed by neurologists (19%), radiation oncologists (11%), general oncologists (3%), and other physicians (7%).

### 3.2. Survey

The questionnaire was completed by 23 physicians: neurosurgeons (n = 7), neurologists (n = 9), and radiation oncologists (n = 7). Table 1, Table 2, Table 3 and Table 4 summarise the survey’s results. Fifty to seventy percent of the referrers rated FET PET as being important or very important for the assessment of newly diagnosed gliomas, including the confirmation of suspected glioma, estimation of tumour extent, biopsy guidance, and planning for surgery and radiotherapy (Table 1). A more detailed analysis revealed differences in the assessment of the various specialist groups for this indication: while approx. 80% of the neurosurgeons and neurologists rated FET PET as important in newly diagnosed gliomas, the rating of the radiation oncologists was only approx. 20% (*p* < 0.01).

In the early postoperative situation, 41% of the physicians considered FET PET helpful. In patients with recurrent glioma, 95% of the referring physicians considered FET PET as important or very important for differentiating tumour recurrence from treatment-related changes (e.g., pseudoprogression and radionecrosis), for the estimation of the recurrent tumour’s extent, and 91% for the planning of additional surgery or radiotherapy. Approximately 50% of physicians rated FET PET as essential for treatment monitoring, especially in glioma patients undergoing chemotherapy. Again, a more detailed analysis revealed differences in the assessment of the various specialist groups for this indication. While 83% of the neurologists rated FET PET as important in this indication, the fraction of neurosurgeons was 43%, and that of the radiotherapists only 20% (*p* < 0.001). In patients with brain metastases, 68% of the physicians considered FET PET important or very important for identifying suspected recurrence, and 50 % for treatment monitoring.

Regarding the fraction of patients for whom FET PET was considered beneficial, the highest values were again achieved for diagnosing recurrent gliomas (68 ± 22%) (Table 2) and was around 50% for other indications. The value of advanced MR procedures for the various indications compared to FET PET was rated as important or very important by approx. 50% of the referrers (Table 3).

In the survey’s final question, 100% of the referring physicians stated that FET PET availability is very important and should be approved for routine use (Table 4). About 90% stated that the availability of FET PET at the FZJ is sufficient for their needs and that FET PET needs to be available in specialised centres only.

## 4. Discussion

The study’s rationale was to obtain information on the clinical acceptance of FET PET in brain tumour diagnosis in an environment that reflects the clinical needs for this examination. The PET facility, which includes a cyclotron and a radiochemistry department, located at the FZJ, is not part of a single university hospital. Instead, it is open to neuro-oncological departments from several university hospitals and specialised centres. Since clinicians are usually under considerable time pressure when assessing the best medical treatment for their patients, they prefer to avoid examinations of little help in the clinical decision-making process. This particularly applies to examinations not available in their clinic. Therefore, there is a high probability that the referral spectrum at our institution reflects the clinical need for FET PET in decisive diagnostic questions, which an MRI did not answer satisfactorily.

First, we would like to emphasise that the considerable increase in the number of FET PET examinations, especially in the last decade, is a strong indicator of the clinical relevance of amino acid PET in brain tumour diagnostics (Figure 1) and the referring clinicians appear to benefit significantly from this information during the clinical decision-making process.

The analysis of the referral spectrum shows that differentiating tumour progression or recurrence from treatment-related changes in patients with glioma is considered the most important indication for FET PET (Figure 2). In recent years, recurrence diagnosis in brain metastases has played an increasing role. Consequently, the diagnosis of tumour recurrence in general accounts for about 59% of all indications. This finding is consistent with a recent study investigating the impact of FET PET/MRI on the clinical management of brain tumour patients [13]. In that study, the proportion of patients with suspected glioma recurrence accounted for 70% of examinations, and FET PET/MRI changed clinical management in 46% of the cases. Accordingly, 95% of all referring physicians in our survey rated FET PET as being important or very important for decision making in recurrent gliomas (Table 1).

The second most common indication for FET PET in our database was the clarification of unclear brain lesions, accounting for 20%. Fifty to seventy per cent of all referring physicians in our survey rated FET PET as important or very important for confirming suspected tumours, as well as for estimating tumour extent and the better planning of biopsy, surgery, or radiation therapy. Brendle et al. reported that FET PET/MRI affected clinical management in 33% of untreated lesions [13]. However, there were significant differences in the rating for this indication among the various specialist groups in our survey. While approx. 80% of neurosurgeons and neurologists rated FET PET as essential in newly diagnosed gliomas, the fraction of radiation therapists was only 20% (*p* < 0.01). This differential assessment may be explained by the fact that the differential diagnosis of newly diagnosed brain lesions concerns neurologists and neurosurgeons more frequently than radiotherapists.

Treatment monitoring also played a crucial role in our collective (19%), but only approx. 50% of the referring physicians rated FET PET as necessary for the clinical decision-making process in this indication. Again, there were significant differences in this indication among the specialist groups. While most neurologists rated FET PET as important for this indication, the fractions of neurosurgeons and radiotherapists were significantly smaller.

It is tempting to speculate that these differences are again related to the differential frequencies with which the different specialists carry out chemotherapies. Although several studies have demonstrated promising results with amino acid PET in treatment monitoring, further studies are warranted to establish this indication in clinical practice [25,26].

In the literature, the significance of FET PET for tumour grading, prognostication, or the prediction of molecular markers is a frequent topic [14,27,28,29,30,31]. Interestingly, this indication was not listed by the referrers in our collective. Our data suggest that tumour classification and prognostication are primarily based on histopathological and molecular evaluation of the tumour specimen. Additional information afforded by FET PET may be beneficial, but does not play a significant role in FET PET indication. A previous study reported that the grading of inhomogeneous masses with predominantly low-grade features and grading in tumour locations at risk for surgery complications accounted for approx. 20% of the FET PET indications in newly diagnosed tumours. This discrepancy may be explained by the specific referral behaviour of the therapists in that centre, but it is also possible that such referral details were not recorded in the patient files of our large long-term collective.

It should be kept in mind that excellent results in solving diagnostic problems in brain tumour imaging can also be achieved by using advanced MR methods, e.g., for differentiating unclear lesion or recurrence diagnosis [1]. However, advanced MRI methods had already been included in the decision process of many patients, suggesting that the referred patients represent a cohort in which diagnosis could not be established satisfactorily by either conventional or advanced MRI. In our survey, approx. 50% of the referrers rated advanced MRI as important for the different indications. Some referrers mentioned that advanced MRI methods were often unavailable due to a lack of appointments. Others indicated that they generally exploit the potential of advanced MR methods before referral for FET PET.

In general, it should be pointed out that interpretating PWI, MRS, or DWI is challenging and requires extensive radiological expertise and experience. In contrast, the interpretation of FET PET findings is relatively easy for the therapist, supporting the method’s acceptance. However, FET PET and advanced MR methods should not be regarded as competing, but complementary. The complementary or additive nature of both approaches is currently the subject of numerous studies with promising results [32,33,34,35,36,37,38].

Given the high clinical relevance of FET PET in brain tumour diagnostics, we would like to comment briefly on the current legal situation. For FET, approval exists in Switzerland [39] and in France (IASOglio, IASON). Clinical use of FET is possible in Germany under the Regulation on Radioactive Drugs or Drugs Treated with Ionizing Radiation (AMRadV), i.e., in centres with a cyclotron and a manufacturing license for the tracer [40]. In December 2021, the Federal Joint Committee approved reimbursement for PET or PET/CT with radioactive amino acids in malignant glial tumours to differentiate post-therapeutic changes from tumour tissue and confirm tumour recurrence as part of the amendment to the guidelines on outpatient specialist care [41]. Thus, a clinical application of FET PET and a reimbursement of the services by the public health insurers is possible in specialised centres. General approval of FET by the German Federal Institute for Drugs and Medical Devices (BfArM) is not yet available.

A limitation of this study is that only physicians who frequently referred patients for FET PET participated in the survey. It cannot be excluded that the opinion of physicians who are critical of FET PET may not have been adequately considered. Therefore, the ratings of the referrers should be considered with caution. On the other hand, all major neuro-oncology centres in the area of the FZJ took part in this survey, and the spectrum of referrers covers all relevant disciplines. Furthermore, as can be seen from the variability of the results, both supporters and sceptics of FET PET are included. The differential assessment within the various indications is independent of this bias and provides relevant information on the preferential indications for FET PET in clinical practice.

## 5. Conclusions

The increasing demand for FET PET, especially to differentiate tumour progression and treatment-related changes in gliomas, and the referring physicians’ judgements provide convincing evidence that this method is relevant in the clinical decision-making process of glioma patients and patients with cerebral metastases. Furthermore, our study strongly supports the need to approve amino acid PET for routine use in brain tumour patients. 

## Figures and Tables

**Figure 1 cancers-14-03336-f001:**
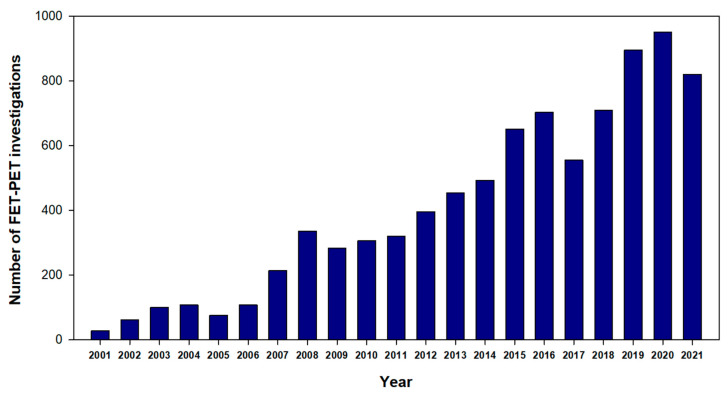
Number of FET PET investigations at the FZJ since 2001.

**Figure 2 cancers-14-03336-f002:**
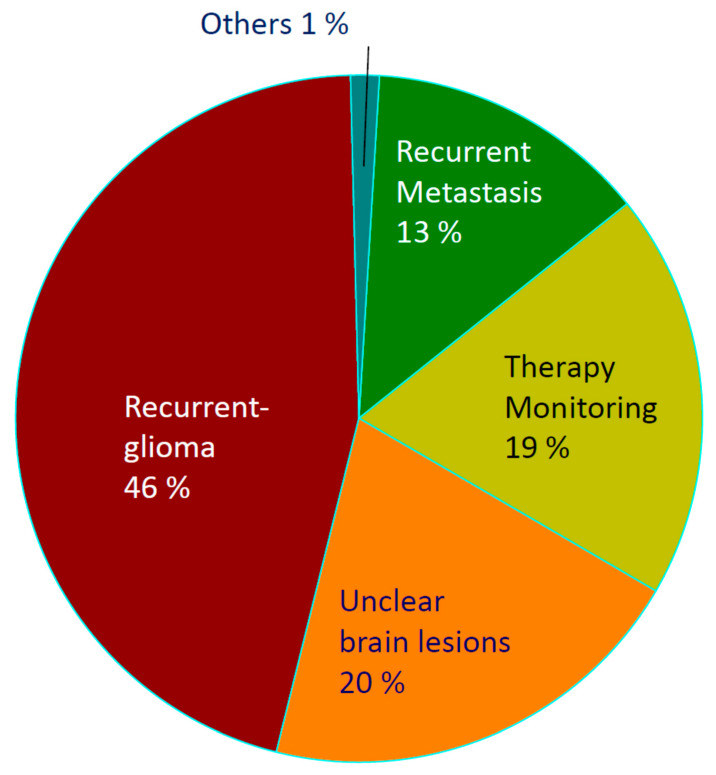
Fraction of indications in 2019.

**Table 1 cancers-14-03336-t001:** Referrers’ rating of the significance of FET PET for different indications.

Indication.	Very Important or Important

**In newly diagnosed gliomas**	
Confirmation of suspected glioma	50%
Extent of glioma	59%
Biopsy guidance	64%
OP/RTX planning	73%

**Early postoperative**	
Tumour residuals (yes/no)	41%
Estimation of tumour extent	45%

**Recurrent gliomas**	
Suspected recurrence (yes/no)	95%
Detection of pseudoprogression/radionecrosis	95%
Extent of the recurrent tumour	95%
Planning of surgery/radiotherapy	91%

**Therapy monitoring in gliomas**	
Temozolomide	55%
PCV scheme	55%
Antiangiogenic therapy	55%
Immunotherapy	50%
Other therapies	53%

**Brain Metastases**	
Suspected recurrence (yes/no)	68%
Therapy monitoring	50%

“Very important or important” = Rating <3 according to a rating scale ranging from 1, very important, to 5, unimportant. Right column: percentage of referrers rating <3.

**Table 2 cancers-14-03336-t002:** Referrers’ rating of the percentage of patients in which FET PET is necessary for different indications.

Indication	Mean ± SD (%)	FET PET Necessary in ≥50% of the Patients
Differential diagnosis of brain lesion	47 ± 31%	55%
Prognosis of gliomas	43 ± 28%	55%
Biopsy guidance in gliomas	50 ± 30%	64%
Tumour extent for OP/RT planning	54 ± 24%	75%
Diagnosis of recurrent gliomas	68 ± 22%	91%
Therapy monitoring in gliomas	53 ± 27%	73%
Diagnosis of recurrent brain metastasis	45 ± 31%	45%
Therapy monitoring of brain metastasis	36 ± 29%	36%

Rating of the referrers regarding the necessity of FET PET in different indications in five steps: <10%; 25%, 50%, 75% and >90% of patients. Right column: percentage of referrers rating FET PET as necessary in ≥ 50% of the patients.

**Table 3 cancers-14-03336-t003:** Rating of the significance of advanced MR procedures (PWI, MRS, and DWI) for different indications compared to FET PET.

Indication	Adv. MRI Very Important or Important
Differential diagnosis of brain lesion	52%
Diagnosis of recurrent gliomas	57%
Therapy monitoring in gliomas	52%
Diagnosis of recurrent brain metastasis	52%
Therapy monitoring of brain metastasis	38%

“Very important or important” = Rating <3 according to a rating scale ranging from 1, very important, to 5, unimportant. Right column: percentage of referrers’ rating <3.

**Table 4 cancers-14-03336-t004:** Referrers’ rating on the general availability of FET PET.

Statement	Correct
The availability of FET PET is very important for me	100%
The availability of FET PET is sufficient for my needs	86%
FET PET should be approved as a standard procedure	100%
FET PET should be available in specialised neuro-oncological centres only	86%

“Correct” = Rating <3 according to a rating scale ranging from 1, correct, to 5, not correct. Right column: percentage of referrers rating <3.

## Data Availability

The data that support the findings of this study are available on request from the corresponding author. The data are not publicly available due to privacy or ethical restrictions.

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
