# Peer review of "Two Decades of Brain Tumour Imaging with O-(2-[18F]fluoroethyl)-L-tyrosine PET: The Forschungszentrum Jülich Experience"

_cancers, 2022, doi:10.3390/cancers14143336_

Round 1
Reviewer 1 Report
This manuscript describes a nice piece of work relating to the clinical acceptance/value of the PET radiopharmaceutical 18FET for imaging brain tumours and the relevance to patient management. Although the tracer or its application is not new, its role in routine clinical applications needs to be promoted. As such, this is a nice progression of the clinical science and would be of value to readers.
Please see comments below
Review FET
This is a nice study that describes the clinical acceptance of an important PET radiopharmaceutical 18FET in the clinical management of patients with brain tumour.
Questions:
1. Opening statement 18FET was developed at FZJ in the 90’s. Is this correct? The reference stated and literature point out that 18FET was developed in Munchen (Wester et. al). Unless I am mistaken- please correct.
2. Under 2.4
Correct ‘Alls’ to ‘All’
3. Under Discussion line 8 change ‘the own clinic’’ to ‘their own clinic.
4. Under discussion paragraph 3 line 3; is it ‘recurrence diagnosis’ or ‘recurrent diagnosis’ ?
General comment:
5. From their survey and clinical network, do the authors know or can comment on what percentage of patients with any brain tumours receive an 18FET scan compared to those that do not? And has the survey suggested as to why the latter patients would not receive an 18F-FET scan?
6. Although it would be difficult without a survey, can the authors comment on the clinical outcomes of patients who had not undertaken an 18F-FET scan compared to those that did?
7. The authors provided a good breakdown on the use, relevance and value of 18FET clinical applications. However, as the authors have stated the source of their survey is from a network of clinical sites surrounding FZJ where 18FET is readily available. Although it is not trivial and to promote the value, acceptance and further approval of 18FET, a comparative study /survey would be highly advantageous if the authors could compare this data to a region(s) where 18FET is not available.
Author Response
We thank the reviever for the valuable comments!
Questions:
- Opening statement 18FET was developed at FZJ in the 90’s. Is this correct? The reference stated and literature point out that 18FET was developed in Munchen (Wester et. al). Unless I am mistaken- please correct.
Reply: H.J. Wester developed FET during his PhD thesis at the FZJ (see Ref. 25 in the cited paper: Wester HJ. N.c.a. F-IR-Fluorination of Proteins. Peptides and Tyrosin. Report no. 3209. Juelich, Germany: Nuclear Research Center Juelich; 1996). After completing his PhD, he moved to Munich as a post-doc and continued to pursue the preclinical testing of FET. We have added the PhD thesis to the reference list to make this point more evident. - Under 2.4 Correct ‘Alls’ to ‘All’
Reply: Done.
- Under Discussion line 8 change ‘the own clinic’’ to ‘their own clinic.
Reply: Done.
- Under discussion paragraph 3 line 3; is it ‘recurrence diagnosis’ or ‘recurrent diagnosis’ ?
Reply: ‘recurrence diagnosis’ is correct, i.e. diagnosis of tumour recurrence
General comment:
- From their survey and clinical network, do the authors know or can comment on what percentage of patients with any brain tumours receive an 18FET scan compared to those that do not? And has the survey suggested as to why the latter patients would not receive an 18F-FET scan?
Reply: Unfortunately, no information is available on what percentage of patients with brain tumours receive an FET scan. However, Table 2 provides the referring physicians' assesment concerning this question. Depending on the clinical indication, FET PET was rated as necessary in 36 to 68% of the patients. Obviously, in the remaining patients, the specific clinical questions could be answered with conventional methods (clinical course, MRI, etc.).
- Although it would be difficult without a survey, can the authors comment on the clinical outcomes of patients who had not undertaken an 18F-FET scan compared to those that did?
Reply: Unfortunately, such information is not available.
- The authors provided a good breakdown on the use, relevance and value of 18FET clinical applications. However, as the authors have stated the source of their survey is from a network of clinical sites surrounding FZJ where 18FET is readily available. Although it is not trivial and to promote the value, acceptance and further approval of 18FET, a comparative study /survey would be highly advantageous if the authors could compare this data to a region(s) where 18FET is not available.
Reply: We agree with this suggestion, but it is beyond our capacity to organize such a survey in a reasonable time. Furthermore, we have doubts about the usefulness of interviewing doctors who have no experience with FET PET on the relevance and role of the method in clinical practice.
Reviewer 2 Report
The authors correctly state that “A limitation of this study is that only physicians who frequently referred patients for FET PET participated in the survey. It cannot be excluded that the opinion of physicians who are critical of FET PET may not have been adequately considered. Therefore, the ratings of the referrers should be considered with caution. On the other hand, all major neuro-oncology centres in the area of the FZJ took part in this survey, and the spectrum of referrers covers all relevant disciplines”. It should be briefly mentioned in the abstract.
Author Response
We thank the reviewer for the valuable comments!
Comments and Suggestions for Authors
The authors correctly state that “A limitation of this study is that only physicians who frequently referred patients for FET PET participated in the survey. It cannot be excluded that the opinion of physicians who are critical of FET PET may not have been adequately considered. Therefore, the ratings of the referrers should be considered with caution. On the other hand, all major neuro-oncology centres in the area of the FZJ took part in this survey, and the spectrum of referrers covers all relevant disciplines”. It should be briefly mentioned in the abstract.
Reply: We agree with this comment and added a sentence concerning the limitations of the study to the introduction.
Reviewer 3 Report
This article performs a survey on specialists to evaluate the relevance of PET PET in the clinical diagnosis of brain tumors. However, the survey was only conducted by specialists who have referred to the use of this technology, which is biased. Also, the sample size and range are pretty limited. Considering the general purpose of this article, the result is not convincing.
Author Response
This article performs a survey on specialists to evaluate the relevance of PET PET in the clinical diagnosis of brain tumors. However, the survey was only conducted by specialists who have referred to the use of this technology, which is biased. Also, the sample size and range are pretty limited. Considering the general purpose of this article, the result is not convincing.
Reply: We have adequately addressed the limitation of the study that only physicians who frequently referred patients for FET PET participated in the survey of this study. On the other hand, it is also not meaningful to interview doctors without experience with FET PET, who cannot assess its role in clinical practice. We disagree with the comment that the sample size and range are pretty limited. Our patient collective is probably the largest FET PET collective available in the world. Given the rarity of the disease and the limited availability of FET PET, the number of specialists surveyed cannot be considered low.
Reviewer 4 Report
The manuscript “Two decades of brain tumour imaging with O-(2-[18F]-fluoro-ethyl)-L-tyrosine PET: The Forschungszentrum Jülich experience” reports interesting results on glioma based on retrospective study and survey anaslyzed database of 6,534 FET PET examinations.
1. Abstract should be rewritten based on findings of the study. Abstracts sounds like introduction/ review of literature. A precise abstract representing concise results and findings would be better.
2. “Therefore, there is a high probability that the referral spectrum at our institution reflects the clinical need for FET PET in decisive diagnostic questions, which MRI did not answer satisfactorily.” Authors must clarify and justify this statement with comparison of those questions which FET PET can answer, and MRI cannot. I do not find a clear discussion on importance of FET PET for glioma and its metastasis diagnosis. It’s great to emphasis that number of FET PET scan increase, however, it’s not the answer to the question that why FET PET is better that MRI scans.
3. “Treatment monitoring also played a crucial role in our collective (19%), but only ap-prox. 50% of the referring physicians rated FET PET as necessary for the clinical decision-making in this indication.” What could be the reason that even 50 % physician do not rate PET scan necessary that should be discussed?
4. “Others indicated that they generally exploit the potential of advanced MR methods before referral for FET PET. It should be pointed out that the interpretation of PWI, MRS, or DWI, is challenging and requires extensive radiological expertise and experience.” This statement contradict the statement mentioned in question 1.
Author Response
We thank the reviewer for the valuable comments!
1.Abstract should be rewritten based on findings of the study. Abstracts sounds like introduction/ review of literature. A precise abstract representing concise results and findings would be better.
Reply: We have rewritten the abstract according to the reviewer’s suggestions.
2.“Therefore, there is a high probability that the referral spectrum at our institution reflects the clinical need for FET PET in decisive diagnostic questions, which MRI did not answer satisfactorily.” Authors must clarify and justify this statement with comparison of those questions, which FET PET can answer, and MRI cannot. I do not find a clear discussion on importance of FET PET for glioma and its metastasis diagnosis. It’s great to emphasis that number of FET PET scan increase, however, it’s not the answer to the question that why FET PET is better that MRI scans.
Reply: In the introduction we had stated that MRI is the method of choice in brain tumour diagnosis, but differentiating tumour tissue from non-specific tissue changes can be problematic. Many studies have demonstrated that PET using radiolabeled amino acids provides decisive additional information to solve these problems. Furthermore, we had stated that several excellent review articles and meta-analyses have discussed the sensitivity, specificity, and diagnostic accuracy of FET PET in various aspects of brain tumour imaging and it is not the intention of the present study to re-review these aspects. Based on this background information, the above statement can be derived conclusively.
3.“Treatment monitoring also played a crucial role in our collective (19%), but only ap-prox. 50% of the referring physicians rated FET PET as necessary for the clinical decision-making in this indication.” What could be the reason that even 50 % physician do not rate PET scan necessary that should be discussed?
Reply: We have discussed this aspect in the following sentences of the discussion:
“Again, there were significant differences in this indication among the specialist groups. While most neurologists rated FET PET as important for this indication, the fractions of neurosurgeons and radiotherapists were significantly smaller. It is tempting to speculate that these differences are again related to differential frequencies with which the different specialists carry out chemotherapies.”
4.“Others indicated that they generally exploit the potential of advanced MR methods before referral for FET PET. It should be pointed out that the interpretation of PWI, MRS, or DWI, is challenging and requires extensive radiological expertise and experience.” This statement contradict the statement mentioned in question 1.
Reply: We agree that the argumentation is misleading and have modified the text accordingly.
Reviewer 5 Report
The authors have reviewed their experience with FET PET since 2005 by studying usage in their institution and surveying referring physicians. They conclude that FET PET is helpful in managing patients with brain tumors and that the studies should be approved for use in this indication. This conclusion is important, particularly if clinicians face push-back from third-party payors when attempting to get approval for the studies. The manuscript is well written. The data speaks for itself, although with some biases, which should be included in the DISCUSSION as limitations or more appropriately as further comment. These potential biases are evident in the following questions which the study did not really address:
1. Does free of charge testing impact referrals for the FET PET? In other words, would clinicians order the test if the patient or their insurance company was required to pay for the study?
2. What was the impact of FET PET on treatment decisions? Could the same decisions have been reached without the study?
3. What percentage of FET PET studies were either unhelpful or confusing to the management of the patient (false positive or false negative)?
One last comment related to the first sentence on the INTRODUCTION, in which the authors may wish to indicate that MRI is first choice for initial diagnosis of brain tumor, as well as to following the disease over time.
Author Response
We thank the reviewer for the valuable comments!
The authors have reviewed their experience with FET PET since 2005 by studying usage in their institution and surveying referring physicians. They conclude that FET PET is helpful in managing patients with brain tumors and that the studies should be approved for use in this indication. This conclusion is important, particularly if clinicians face push-back from third-party payors when attempting to get approval for the studies. The manuscript is well written. The data speaks for itself, although with some biases, which should be included in the DISCUSSION as limitations or more appropriately as further comment. These potential biases are evident in the following questions which the study did not really address:
- Does free of charge testing impact referrals for the FET PET? In other words, would clinicians order the test if the patient or their insurance company was required to pay for the study?
Reply: The free availability of FET PET is an important aspect of our study and certainly influences the use of the procedure. We know that neurooncologists at other university hospitals avoid FET PET because of internal billing, but this also applies to other tests such as advanced MRI. However, this does not apply to reimbursement by health insurance. The health insurace companies reimburse about 90% of the FET examinations in the FZJ.
- What was the impact of FET PET on treatment decisions? Could the same decisions have been reached without the study?
Reply: Our study did not examine this aspect because such data was not available in the patient files. We refer to other studies in this regard (see Ref. 12)
- What percentage of FET PET studies were either unhelpful or confusing to the management of the patient (false positive or false negative)?
Reply: See comment above. This aspect was not examined in our study because such data was not available in the patient files. We refer to other studies in this regard (Ref. 12)
One last comment related to the first sentence on the INTRODUCTION, in which the authors may wish to indicate that MRI is first choice for initial diagnosis of brain tumor, as well as to following the disease over time.
Reply: We have added this aspect to the introduction.
Round 2
Reviewer 4 Report
The authors revised the manuscript based on reviewers comments, however, the response to reviewer's comments are not up to scratch. Because this is a review article therefore I would recommend for the acceptance in peer reviewed journal Cancers for the interest of readers.